# Differences in the Residual Behavior of a Bumetrizole-Type Ultraviolet Light Absorber during the Degradation of Various Polymers

**DOI:** 10.3390/polym16020293

**Published:** 2024-01-21

**Authors:** Hisayuki Nakatani, Taishi Uchiyama, Suguru Motokucho, Anh Thi Ngoc Dao, Hee-Jin Kim, Mitsuharu Yagi, Yusaku Kyozuka

**Affiliations:** 1Polymeri Materials Laboratory, Chemistry and Materials Engineering Program, Nagasaki University, 1-14 Bunkyo-machi, Nagasaki 852-8521, Japan; bb52122606@ms.nagasaki-u.ac.jp (T.U.); motoku@nagasaki-u.ac.jp (S.M.); anh.dao@nagasaki-u.ac.jp (A.T.N.D.); 2Organization for Marine Science and Technology, Nagasaki University, 1-14 Bunkyo-machi, Nagasaki 852-8521, Japan; kyozuka@nagasaki-u.ac.jp; 3Graduate School of Fisheries and Environmental Sciences, Nagasaki University, 1-14 Bunkyo-machi, Nagasaki 852-8521, Japan; heejin@nagasaki-u.ac.jp (H.-J.K.); yagi-m@nagasaki-u.ac.jp (M.Y.)

**Keywords:** microplastics, ultraviolet light absorber, bleaching, polypropylene, polystyrene, polyethylene

## Abstract

The alteration of an ultraviolet light absorber (UVA: UV-326) in polymers (PP, HDPE, LDPE, PLA, and PS) over time during degradation was studied using an enhanced degradation method (EDM) involving sulfate ion radicals in seawater. The EDM was employed to homogeneously degrade the entire polymer samples containing the UVA. The PP and PS samples containing 5-phr (phr: per hundred resin) UVA films underwent rapid whitening, characterized by the formation of numerous grooves or crushed particles. Notably, the UVA loss rate in PS, with the higher glass transition temperature (Tg), was considerably slower. The behavior of crystalline polymers, with the exception of PS, was analogous in terms of the change in UVA loss rate over the course of degradation. The significant increase in the initial loss rate observed during EDM degradation was due to microplasticization. A similar increase in microplasticization rate occurred with PS; however, the intermolecular interaction between UVA and PS did not result in as pronounced an increase in loss rate as observed in other polymers. Importantly, the chemical structure of UVA remained unaltered during EDM degradation. These findings revealed that the primary cause of UVA loss was leaching from the polymer matrix.

## 1. Introduction

Plastic debris has been discharged into the ocean, resulting in the widespread distribution of microplastic (MP) pollution [1,2,3,4,5,6,7,8]. Reflecting the substantial production volume of commercial plastics, MPs are primarily comprised of polyethylene (PE), polypropylene (PP), and polystyrene (PS) particles that float on the sea surface. Given their non-biodegradable nature, these polymeric products persist in the marine environment for extended periods. Initial investigations proposed the transfer of chemicals from plastic debris ingested by sea birds to their tissues [9]. The evaluation of chemical transfer from marine plastic debris such as MPs to other organisms in the oligotrophic open ocean is yet to be elucidated. However, due to the size of micrometers, MPs are deemed bioavailable across the entire food web, calling for a consideration of chemical transfer at the bottom of the food chain where bioaccumulation could potentially initiate.

MPs can form in diverse locations, with some generated in the terrestrial region (on land) and others in the sea. Occasionally, MPs partially transition from the sea to the atmosphere, circulating in various places. Halle et al. outlined the mechanism of MP formation in the sea, linking it to sunlight exposure [4]. A total of 92% of the collected samples in the study were made of PE, the type of polymer that is principally degraded by photo-oxidation, in which UV radiation triggers free radical reactions in the polymeric chain [10]. In studying photo-oxidation degradation, the broadening of the carbonyl groups (measured by Fourier transform infrared spectroscopy, or FT-IR), particularly the ketone, carboxylic acid, and ester functional groups formed during UV irradiation, is commonly used to assess the progression of weathering within the 1650−1850 cm^−1^ interval. The carbonyl index (or CI) obtained from such measurements works as a metric for characterizing the advancement of polymer oxidation. The MPs in Halle’s study exhibited carbonyl indexes ranging from 0.13 to 0.74, signifying advanced stages of oxidation. While the detailed mechanism for fragmentation remained unclear, it was evident that sunlight, especially in the visible and/or UV regions, played a role in MP formation.

The substantial presence of large quantities of MPs raises concerns about potential risks to marine ecosystems. However, the quantitative impact of MPs on marine life remains uncertain, and there is no definite answer to the numerous concerns surrounding this issue. Plastic products, which are rarely used alone and often contain various types of organic or inorganic compounds, and among them, organic additives [11,12,13,14], are of particular concerns. These additives may undergo transformation during MP formation. Recently, Z. Tian et al. reported the induction of acute mortality in coho salmon by a highly toxic quinone transformation product of N-(1,3-dimethylbutyl)-N′-phenyl-p-phenylenediamine (6PPD), a globally ubiquitous tire rubber antioxidant [15]. The toxicity of 6PPD-quinone to juvenile coho salmon was significant, with a median lethal concentration (LC_50_) of 0.79 ± 0.16 μg/liter. Throughout the product life cycle, antioxidants such as 6PPD are specifically engineered to diffuse to tire rubber surfaces. Their primary function is to scavenge atmospheric and ground-level ozone, creating protective films that eliminate the ozone-mediated oxidation of rubber elastomers [16]. Consequently, the intentional reaction of 6PPD in tire rubbers leads to the formation of 6PPD-quinone and related transformation products, inadvertently elevating toxicity and environmental risk.

In the study of the ecological impact of MPs, critical questions arise: How much of the additives leach out during MP formation? What changes occur in the chemical structures of additives? Additionally, how does the additive behavior of different plastics vary during MP formation? The above-mentioned case of 6PPD highlights the transformation of a seemingly safe additive into a toxic compound in nature, emphasizing the importance of understanding chemical changes during MP formation for assessing its impact on living organisms. Notably, UV stabilizers (UVAs), essential additives in plastic substances, particularly for outdoor packaging and corrosion inhibition, warrant detailed investigation after the degradation of containing plastics. Bumetrizole-type UVA, frequently used in commodity plastics such as PP and PE, shares a chemical structure similarity with 6PPD, and its degraded product forms a quinone body akin to 6PPD. Thoroughly investigating UVA changes post degradation is essential due to potential ecological impacts.

As mentioned above, the primary locations of MP formation are considerably complicated, occurring both on land and in the sea, and each is governed by distinct mechanisms [17]. Surface observations of MPs retrieved from the seashore reveal delamination, whereas those from the riverside exhibit an abrasion patch structure [17]. The delaminated part, releasing smaller MPs into the sea, is a crucial aspect. Although MP formation occurs both on land and in the sea, it is considered to primarily be produced in the sea and its vicinity. Consequently, reference samples formed in the sea are deemed more suitable for investigating the leaching and alteration behavior of UVA during MP formation. Delamination, achieved by exposure to visible and/or UV light in water, has been utilized by several researchers to obtain MP reference samples [18,19,20]. Similarly, in our previous study, we conducted a photodegradation test of PP film in water under visible light irradiation using a specific photocatalyst, resulting in planar delamination and acquisition of the MP reference sample [21]. While MP formation is inherently linked to autoxidation, the rates induced by light irradiation, even with a photocatalyst, were notably sluggish. To facilitate a comparative study of additive behavior on different plastics during MP formation, the development of an accelerated MP production method is imperative [22,23]. In our previous study [24], we degraded PP in seawater using a sulfate ion radical (SO_4_•^−^) as a highly efficient initiator for plastic degradation. This combination, coupled with pH control, significantly accelerated the degradation process. Notably, this approach was effective for the accelerated degradation of other C-C bonded polymers such as HDPE, LDPE, PLA, and PS, which share the same degradation mechanism (autoxidation).

In this study, we studied the quality-altering behavior of bumetrizole-type UVA (UV-326) in PP, HDPE, LDPE, PLA, and PS over time during degradation using an enhanced degradation method (EDM) employing a sulfate ion radical in seawater. UVA concentration in these polymers was quantified through pyrolysis gas chromatography/mass spectrometry (Py-GC/MS), and structural changes in UVA were confirmed through FT-IR measurements.

## 2. Materials and Methods

### 2.1. Materials

PP was supplied by Prime Polymer Co., Ltd., Tokyo, Japan (product name: J-700GP). The melt flow rate (MFR) and density were 8 g/10 min and 0.9 g/cm^3^. PLA was supplied by Mitsui Chemicals, Inc., Tokyo, Japan (production name: Gread H-100). The weight-average molecular weight (Mw) and molecular weight distribution (Mw/Mn) were 1.2 × 10^5^ and 1.1, respectively. PS was purchased from Sigma-Aldrich Co. LLC(St. Louis, MO, USA) The weight-average molecular weight (Mw) and molecular weight distribution (Mw/Mn) were 3.5 × 10^5^ and 2.1, respectively. HDPE and LDPE were purchased from Sigma-Aldrich Co. LLC. The melt index (190 °C/2.16 kg) values of HDPE and LDPE were 10 g/10 min and 25 g/10 min, respectively. Bumetrizole (UVA: UV-326) and potassium persulfate (K_2_S_2_O_8_) were purchased from Wako Pure Chemical Industries (Osaka, Japan) Seawater was prepared with Gex artificial saltwater purchased from Amazon.co.jp.

### 2.2. Preparation of Polymer Samples Containing Ultraviolet Light Absorber (UVA)

The polymer and UVA blend were prepared using an Imoto Seisakusyo IMC-1884 melting mixer (Kyoto, Japan). The polymer pellets (ca. ϕ5 mm) and bumetrizole fine particles (ca. 0.01 mm × 0.1 mm) were employed as polymer and UVA, respectively. Next, 2 g-powdery of polymer sample and 0.1 g UVA were put into a 50 mL glass vessel and then premixed. The mixture was put into the melting mixer, and melt mixing was performed at 180 °C and 50 rpm for 6 min. The obtained sample was molded into a film (ca. 50 × 50 × 0.075 mm) by compression molding at 180 °C under 10 MPa for 5 min. The film was cut to 5 mm × 5 mm × 0.075 mm in size and was employed as polymer film containing 5-phr (phr: per hundred resin) UVA. For PS samples, 10-phr UVA-containing films were prepared in a similar method. 

### 2.3. Degradation Using Sulfate Ion Radicals in Pure Water

The polymer film samples, in their as-prepared state, were employed for the degradation procedure, following the methodology outlined in our previously published report [22]. In this process, five pieces of each film sample were put into individual 100 mL glass vessels, each containing a 20 mL aqueous solution with 0.54 g K_2_S_2_O_8_. The vessels were maintained at ca. 65 °C for a duration of 12 h, with continuous stirring at a tip speed of ca. 100 rpm. To account for the consumption of the oxidant, an equivalent amount of K_2_S_2_O_8_ aqueous solution was renewed every 12 h. This treatment was repeated for various time periods, subjecting the films to a continuous degradation process. The degradation was performed in pure water using sulfate ion radicals.

### 2.4. Degradation Using Sulfate Ion Radicals in Seawater (Enhanced Degradation Method)

The degradation was performed in seawater using a sulfate ion radical. The procedure was according to our previous report [21]. (1) Five pieces of each film were put into a 100 mL glass vessel containing a 20 mL of seawater solution with 0.54 g K_2_S_2_O_8_ at ca. 65 °C for 12 h under stirring with a stirrer tip speed of ca. 100 rpm. (2) An equal amount of K_2_S_2_O_8_ seawater solution was added to compensate for the consumption of oxidant, and its degradation was carried out for 12 h under the same conditions. (3) The five pieces of the film were then transferred to a new 100 mL glass vessel containing 20 mL of seawater solution with 0.54 g K_2_S_2_O_8_, and the degradations were started again under the same conditions. The enhanced degradation method was carried out for a predetermined number of 15 days using (1) to (3) as one set (total 15 sets). The pH value of the solution was changed from 8.2 to 3 during each set. 

### 2.5. Fourier Transform Infrared (FT-IR) Analysis

For the IR spectra, 16 scans were measured with an FT-IR spectrometer (Jasco FT-IR 660 plus, Tokyo, Japan) at a resolution of 4 cm^−1^ over the full mid-IR range (400–4000 cm^−1^). The carbonyl index (CI) of PP was calculated as the band intensity ratio of the carbonyl group (ca. 1715 cm^−1^)/scissoring CH_2_ group (ca. 1450 cm^−1^). The presence of hydroperoxide was confirmed by the development of a broad peak around 3400 cm^−1^.

### 2.6. Pyrolysis Gas Chromatography/Mass (Py-GC/MS) Spectroscopy Measurement and Creation of Calibration Curve

A multi-functional pyrolyzer (Frontier Labs., Fukusima, Japan, EGA/PY-3030D) was attached to a GC/MS spectroscope (SHIMADZU, Kyoto, Japan, GCMS-QP2010 PLUS). The measurement was performed on a 100 μg sample. The pyrolysis was performed at 550 °C. Helium was used as the carrier gas for the capillary column with a flow rate of 1.0 mL/min. The MS system was operated under electron ionization mode at 70 eV.

The UVA (UV-326) concentration in various polymers was measured by the Py-GC/MS. The UVA (UV-326) peaks were identified and assigned in the Py-GC/MS by comparing them to the UVA-free polymer samples. As shown in Appendix A, the UVA peak appeared at a retention time of 23.5 to 24.5 min. The area ratio calculated by dividing the peak area by the total area was plotted against known UVA concentrations to create a calibration curve (see Appendix A). The UVA concentrations in various polymers at the time of degradation were calculated using this calibration curve based on area ratios obtained from the py-GC/MS measurements.

### 2.7. Scanning Electron Microscope (SEM) with Energy Dispersive X-Ray Spectroscopy Analysis

The SEM/EDX analysis was carried out with a JSM-7500FAM (JEOL, Tokyo, Japan) at 5.0 kV. The working distance was about 3 × 4 mm. Samples were placed in a drying oven maintained at 27 °C for 30 min and were sputter-coated with gold before SEM imaging.

### 2.8. Gel Permeation Chromatography (GPC) Analysis

PP samples were dissolved in 5ml of o-dichlorobenzene, and their molecular weights were directly measured using an HLC-8321GPC/HT GPC system (Tosoh Co., Ltd., Tokyo, Japan) at 140 °C.

PS samples were dissolved in 5 mL of chloroform in a small vial, and the obtained sample solution was directly measured by GPC. The molecular weight was determined by GPC (SHIMADZU, Kyoto, Japan, Prominence GPC system) at 40 °C using chloroform as a solvent. 

## 3. Results and Discussion

### 3.1. Homogeneous Degradation Behavior

Weathering degradation primarily affects the polymer matrix’s surface, making it a prolonged process for degradation to reach the interior. Analyzing the quality change of an encapsulated UVA additive within the polymer over a short timeframe is challenging. To facilitate the detection of UVA quality changes during degradation, a method that allows for simultaneous initiation on the surface and within the polymer matrix is essential. We employed a novel “enhanced degradation method (EDM)” using sulfate ion radicals in seawater to uniformly degrade the entire sample containing a UVA (UV-326). EDM has proven effective in homogeneously degrading polymers such as polypropylene in a short duration [24]. The sulfate ion radical (SO_4_•^−^) was chosen as the initiator due to its conversion to hypochlorite (ClOH) upon reaction with chlorine ion in seawater. ClOH, with an extended lifetime, penetrates deep into the polymer matrix before dissociating into radicals and initiating degradation (autoxidation). This approach ensures simultaneous degradation progression from the polymer’s interior and surface, synergistically accelerating MP formation. SO_4_•^−^ generates OH• and ClOH, overcoming inhibitory effects and expediting the degradation process. Figure 1 illustrates the schematic representation of simultaneous and uniform degradation progression in the polymer matrix using the EDM. The behavior of UVA in each polymer during degradation was intricately examined using the EDM.

Figure 2 and Figure 3 depict the color changes in PP and PS samples containing 5-phr UVA film using the EDM, respectively. Samples degraded with the EDM exhibited more pronounced whitening than those degraded with a degradation initiator combining sulphate radicals and pure water. SEM images after 15 days of EDM degradation revealed numerous grooves or crushed particles. In pure water, radical species cannot diffuse into the interior of the polymer, restricting degradation to the surface, similar to weathering degradation. In addition, minimal surface changes indicate a considerably slower degradation rate compared to EDM. Notably, the difference between PP and PS lies in the whitening rate during EDM degradation. Chemi-crystallization, occurring when a polymer chain of semi-crystalline polymers such as PP is broken due to degradation [25], causes volume shrinkage and the observed grooves in SEM photographs of PP samples. It is well-known that semi-crystalline polymers such as PP and HDPE undergo morphological changes, including increased crystallinity and whitening, as the degradation progresses [25,26]. The higher whitening rate in the PP sample is attributed to chemi-crystallization, while the amorphous nature of PS does not undergo such chemical crystallization, resulting in a slower whitening rate. 

### 3.2. UVA Bleaching Behavior

Appendix A shows the degradation time dependency of residual UVA in various polymer samples using sulfate ion radicals in pure water. The order of residual UVA amounts in each polymer, with increasing degradation time, was as follows: PS > PLA > HDPE> PP > LDPE. This order appeared to correlate with the glass transition temperature (Tg) except for PP and HDPE [27]. The UVA loss rate for each polymer was calculated by differentiating the residual amount concerning degradation time, as summarized in Appendix A. PS, with the highest Tg among these polymers, exhibited a UVA loss rate close to zero, independent of degradation time. Other polymers displayed similar behaviors in the change in the UVA loss rate concerning degradation time. The rate reached a maximum within the initial 3 days of degradation and sharply dropped to nearly zero after 6 days, with some variation. The scattering degree seemed linked to crystallinity levels because it was much lower for PS and LDPE with no or low crystallinity. The UVA loss behavior during progressive degradation would be closely associated with mobility of polymer chain. Higher polymer chain mobility, resulting in a lower Tg as the polymer chain easily transitions from a glassy state, indicates more substantial degradation. The presence of a crystalline part hinders the propagation of degradation and the movement of polymer chains. Therefore, if the influence of the crystalline part can be suppressed, the correlation between the UVA loss rate and Tg—in other words, the mobility of the polymer chain—becomes clearer.

Figure 4 and Figure 5 show the degradation time dependence of UVA residual amount and loss rates in various polymer samples using the EDM. Although polymer degradation is commonly limited to the amorphous part, the EDM exhibits the capability to degrade both crystalline and amorphous components [24]. As shown in Appendix A, molecular weight curves of PP and PS samples shifted towards the lower molecular weight with increasing degradation time while maintaining a consistent shape. The reduction behavior of molecular weight was similarly observed in samples of other polymers. The uniformity in molecular weight reduction suggests that EDM-induced degradation progressed uniformly with minimal distinction between crystalline and amorphous parts. In fact, for the crystalline polymers, such as PP, HDPE, LDPE, and PLA, the variation in UVA residuals at different degradation times significantly improved (see Figure 4). Figure 5 further demonstrates the degradation time dependence of UVA loss rate in these crystalline polymers, showing remarkable similarity. Appendix A provides a comparison of degradation time dependence of UVA residual in PP samples, revealing differences between degradation in pure water, sulfate ion radical in pure water, and EDM. The equilibrium of UVA leaching from the polymer matrix can be explained by a physicochemical model in the absence of degradation. The leaching behavior of several UV stabilizers, including UV-326, from various polymer matrices has already been studied by Do et al. [28]. According to their report, the leaching behavior of UVA from LDPE in an acrylic nitrile/water mixture showed a positive deviation from Raoult’s law and negative deviation for PS. Since PP is an aliphatic hydrocarbon polymer as well as LDPE, its UVA deviation value can be regarded as nearly equal to that of LDPE. The positive deviation means that there is no intermolecular interaction between PP and UVA, while the negative value for PS indicates the presence of one [28]. The significant increase in the initial loss rate observed during the EDM-induced degradation in crystalline polymers such as PP, HDPE, LDPE, and PLA is attributed to homogeneous and rapid fragmentation of the polymer matrix, leading to an increased surface area due to microplasticization [24]. A similar increase in microplasticization rate occurred in PS, but the intermolecular interaction between UVA and PS resulted in a less pronounced increase in loss rate compared to other polymers. 

Figure 6 illustrates the scheme of UVA (UV-326) destabilization through autoxidation. The hindered phenol segment of UVA-326 acts as an antioxidant [29,30]. Polymers with aliphatic main chains, such as PP and PS, undergo degradation via autoxidation [31,32], leading to the generation of peroxide radicals (ROO•) from the aliphatic polymer main chains. As shown in Figure 6, the phenol part in the UVA reacted with the ROO•, and hydrogen transfer transpired from the phenol to the peroxide radical [29]. The resulting transformation product, quinone, was formed by reaction with hydroperoxide via the intermediate phenoxyl radical, as shown in Figure 6. The quinone part is an unstable compound and contributes to further alteration, ultimately leading to the decomposition of UVA. This decomposition process requires contact with the peroxide radical, a factor influenced by the polymer chain mobility. UV-326, possessing an aromatic ring, interacted with the benzene ring of PS through π-π stacking. If some of the aromatic rings are converted to unstable quinone rings due to UVA autoxidation, as shown in Figure 6, the interaction would decrease, thereby increasing the UVA leaching rate. Figure 7 shows the FT-IR spectra of 0- and 15-day degraded PP samples using the EDM are similar to those of the hydroperoxide group. In the IR spectrum of the 15-day degraded PP sample, a hydroperoxide-derived peak is observed around 3400 cm^−1^, confirming that PP is degraded by the EDM. In addition, the CI value of 15-day degraded PP increased to approximately 0.6, which also confirms that the degradation reaction (autoxidation) progressed with increasing degradation time, as shown in Figure 8. Appendix A show the IR spectra changes of PP containing 5-phr UVA and PS containing 10-phr UVA films, respectively, degraded using the EDM. The quinone group in benzophenones exhibited a vibrational mode in the spectral region of 1650–1680 cm^−1^ in FT-IR measurements [22,33,34]. After 15 days of degradation of PP containing 5-phr of UVA with the EDM (see Appendix A), peaks assigned to quinone groups were not observed in the relevant spectral region. This result supports the assumption mentioned above that the loss of UVA additive was not due to quinone deterioration but to increased solutes from outside the system, due to the increased surface area resulting from matrix degradation-induced fragmentation. HDPE, LDPE, and PLA, which are crystalline and do not interact with the UVA, are similar to PP in that the UVA loss during EDM-induced degradation is not due to deterioration but to an increased leaching rate associated with fragmentation. Even when degradation using the EDM was performed by increasing the UVA dosage in PS, which exhibited a slow UVA leaching rate, no quinone group peaks appeared until 24 days of degradation, as shown in Appendix A. These results indicate that UVA (UV-326) deterioration in the polymers does not or rarely occurs with degradation, and the primary cause for the loss is leaching out of the system.

## 4. Conclusions

To elucidate the evolving behavior of UVA over time during degradation, we employed EDM to homogeneously degrade the entire polymer sample containing it. The PP and PS samples containing 5-phr UVA films underwent rapid whitening, and their SEM photographs revealed numerous grooves or crushed particles after 15 days of EDM degradation. Notably, there was a difference in the whitening rate between PP and PS during EDM degradation. Chemi-crystallization occurred in semi-crystalline polymers such as PP, causing volume shrinkage and the generation of grooves during degradation. The higher whitening rate observed in the PP sample was attributed to the degradation caused by chemi-crystallization. The UVA loss rate in PS, with the higher Tg, was considerably slower compared to the crystalline polymers (PP, HDPE, LDPE, and PLA). Except for PS, the crystalline polymers exhibited similar behavior in the change in UVA loss rate with respect to degradation time. The significant increase in the initial loss rate observed during the EDM degradation of crystalline polymers was attributed to homogeneous and faster fragmentation of the polymer matrix, leading to increased surface area due to microplastics formation. A similar increase in microplasticization rate was observed in PS, although the intermolecular interaction between UVA and PS did not result in as pronounced an increase in loss rate as observed in other polymers. The UVA (UV-326) has one aromatic ring, which interacted with the benzene ring of PS through π-π stacking. The interaction caused a decrease in the UVA leaching rate. The phenolic part of the UVA was able to react with the degraded polymer, likely resulting in the formation of a quinone compound. As a result, the UVA may undergo degradation, possibly leading to a decrease in residual amounts, a factor other than leaching. However, no quinone group peaks were observed in the FT-IR measurements of EDM degradation of PP and PS containing UVA. These results indicate that the UVA remained unaltered or underwent minimal changes within the polymers. This study concludes that leaching from the system was the primary cause of the loss.

## Figures and Tables

**Figure 1 polymers-16-00293-f001:**
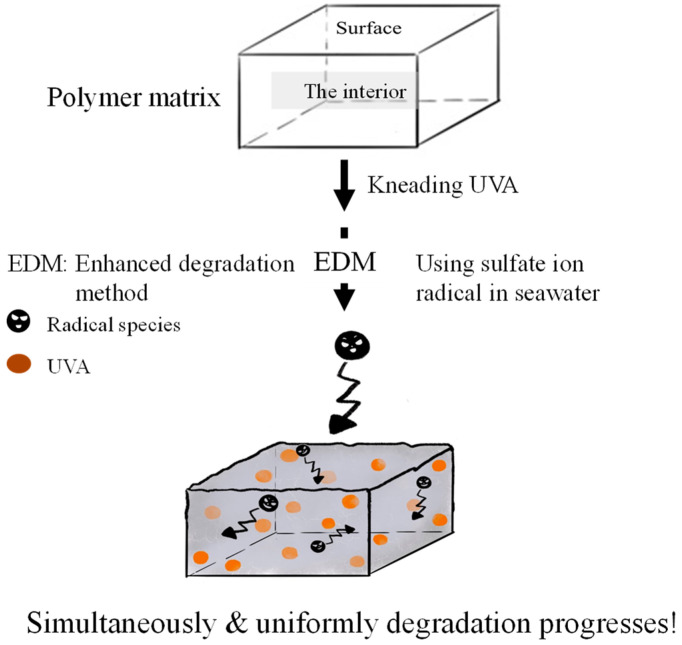
Schematic scheme of simultaneous and uniform degradation progression in polymer matrix using EDM.

**Figure 2 polymers-16-00293-f002:**
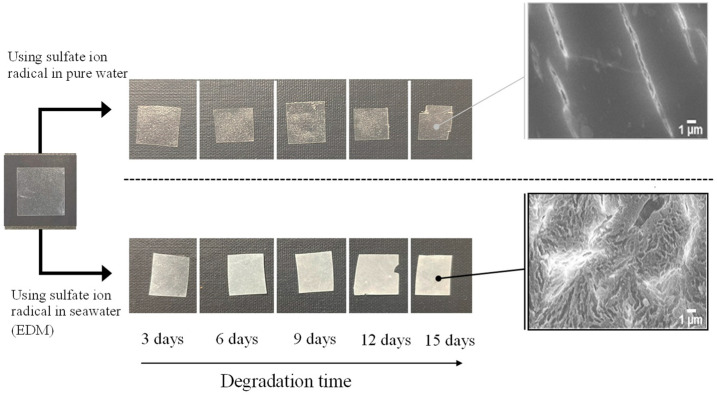
Color change of PP containing 5-phr UVA film using sulfate ion radicals in pure water and in seawater (EDM).

**Figure 3 polymers-16-00293-f003:**
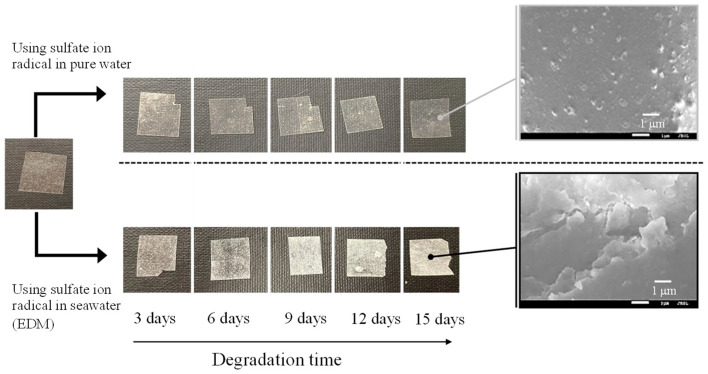
Color change of PS containing 5-phr UVA film using sulfate ion radicals in pure water and in seawater (EDM).

**Figure 4 polymers-16-00293-f004:**
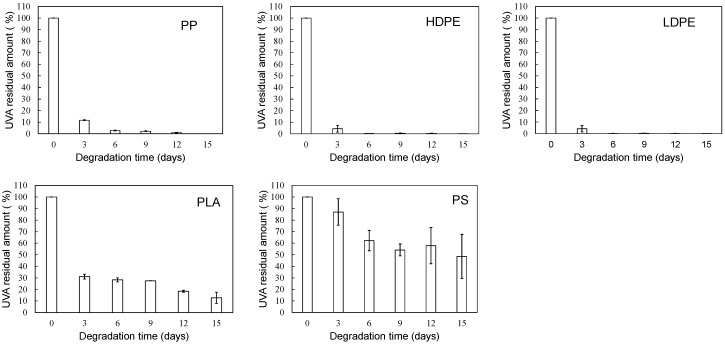
Degradation time dependence of UVA residual amount in various polymer samples using sulfate ion radical in seawater (EDM).

**Figure 5 polymers-16-00293-f005:**
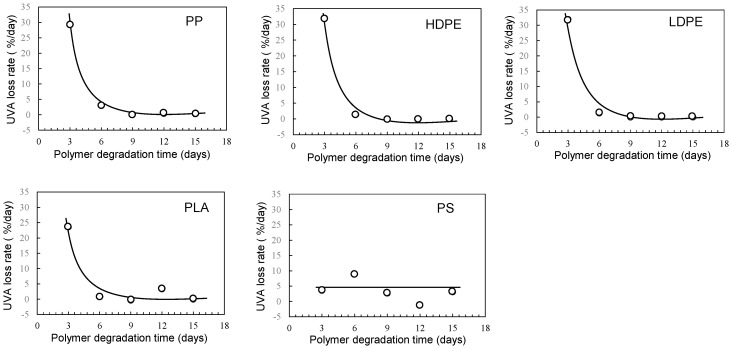
Degradation time dependence of UVA loss rates in various polymer samples using sulfate ion radical in seawater (EDM).

**Figure 6 polymers-16-00293-f006:**
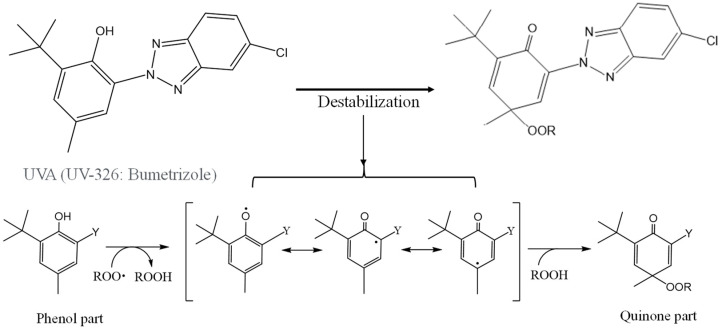
Schematical of UVA (UV-326) destabilization by autoxidation.

**Figure 7 polymers-16-00293-f007:**
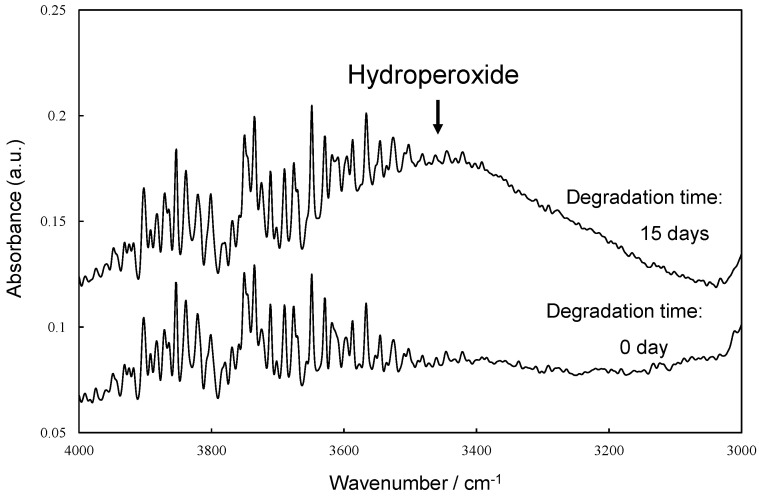
FTIR spectra of 0 and 15 days degraded PP samples by EDM at around hydroperoxide group.

**Figure 8 polymers-16-00293-f008:**
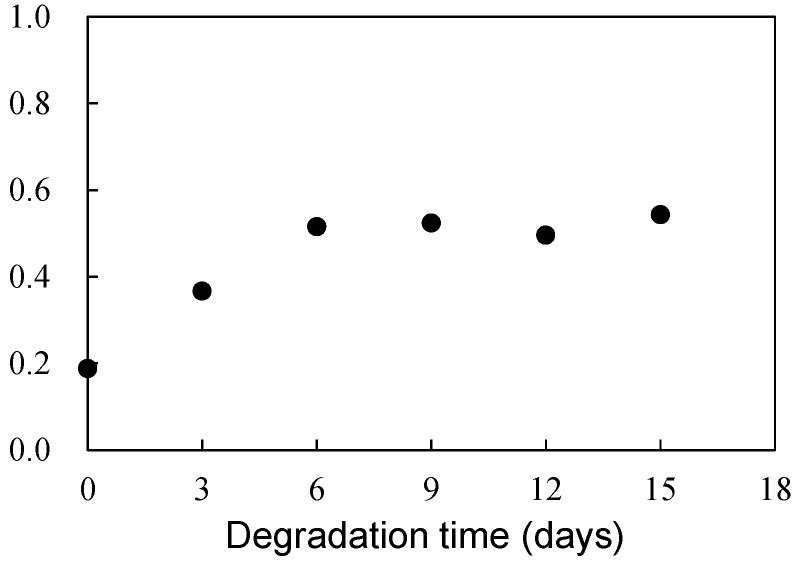
Degradation time dependence of carbonyl index (CI) in PP sample degraded by EDM.

## Data Availability

Data are contained within the article.

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
