# Peer review of "Differences in the Residual Behavior of a Bumetrizole-Type Ultraviolet Light Absorber during the Degradation of Various Polymers"

_polymers, 2024, doi:10.3390/polym16020293_

Round 1

Reviewer 1 Report

Comments and Suggestions for Authors

The manuscript by Nakatani et al., “Differences in the residual behavior of a bumetrizole type ultraviolet light absorber during formation of various microplastics” describes a methodology to follow the degradation of different plastics in seawater using potassium persulfate. The title of the manuscript leads one to believe that the formation of microplastics was the intent of this work, however, this was not the case. The title should better reflect the content of the manuscript.

The manuscript is of some interest to the wider scientific community, and is generally well written, however there are several issues that must be addressed before this manuscript can be considered for publication.

Inconsistent data for the obtained polymer samples is provided. Some samples have molecular weights and dispersity values provided, others do not. Was this data provided by the supplier? Were these values determined by the authors?

FTIR analysis was used to determine CI values. There is no CI data provided, nor is a discussion of CI values provided in the manuscript. Remove this text or provide some discussion in the manuscript.

There is no indication that any control reactions were performed. The main text indicates that experiments were conducted with pure water. These methods also need to be described and/or the reference provided.

 How were the peaks for UVA identified and assigned in the Py-GC/MS? Was Py-GC/MS conducted on UVA-free polymer samples? Reference?

The data in Figures 4 & 5 and Figures 6 & 7 can be combined. The graphs present similar trends; why are both analyses needed? Some of this data can be moved to the ESI file.

The fitted lines in Figures 5 and 7 are not overly representative of the raw data for some of the polymer samples. Error bars should be included.

The interaction between UVA and the PS sample is speculative. Can evidence/reference be provided.

There are two different references labelled as Reference 1. This makes it difficult to determine if the in-text references are correct/relevant.

References 10 and 11 are the same reference.

Figure S2 – The linear regression lines extend beyond the presented data for PP and PS samples. There is no indication in the Methods and Materials section that PP samples were prepared with 10-phr UVA.

Figure S3 shows the changes in MW distribution for PP samples degraded in pure water and sea water. A procedure for determining MW should be included.

Minor issues that need to be addressed:

 Line 91 – Define the abbreviation phr.

Line 95 – Confirm that these are the correct references. Ref 16 seems more appropriate than Ref 26.

Line 126 – “Samples were placed in dried oven...” should be clarified. Drying oven?

Line 177-178 – “…can also degrade the crystalline it.” Must be corrected. Crystallinity?

Line 187 – “ If there is no degradation of UVA due to degradation...” This statement reads awkwardly.

Line 201 – “as pronounced its increase” would be better stated as “as pronounced an increase”

Comments on the Quality of English Language

There are only minor grammatical mistakes. See comments regarding lines 126, 177-178, 187, and 201. There are other issues that affect the readability of the manuscript such as tense shifts, incorrect verb forms, verb agreement, etc.

Author Response

The manuscript by Nakatani et al., “Differences in the residual behavior of a bumetrizole type ultraviolet light absorber during formation of various microplastics” describes a methodology to follow the degradation of different plastics in seawater using potassium persulfate. The title of the manuscript leads one to believe that the formation of microplastics was the intent of this work, however, this was not the case. The title should better reflect the content of the manuscript.

Answer: We revised our title.

The manuscript is of some interest to the wider scientific community, and is generally well written, however there are several issues that must be addressed before this manuscript can be considered for publication.

Inconsistent data for the obtained polymer samples is provided. Some samples have molecular weights and dispersity values provided, others do not. Was this data provided by the supplier? Were these values determined by the authors?

Answer: The data was provided by the supplier.

FTIR analysis was used to determine CI values. There is no CI data provided, nor is a discussion of CI values provided in the manuscript. Remove this text or provide some discussion in the manuscript.

Answer: We added CI data and the corresponding sentences in Results and Discussion.

There is no indication that any control reactions were performed. The main text indicates that experiments were conducted with pure water. These methods also need to be described and/or the reference provided.

Answer: We added the experiment with pure water as “2.3 Degradation using sulfate ion radicals in pure water”.

 How were the peaks for UVA identified and assigned in the Py-GC/MS? Was Py-GC/MS conducted on UVA-free polymer samples? Reference?

Answer: We added “The UVA (UV-326) peaks were identified and assigned in the Py-GC/MS by comparing them to the UVA-free polymer samples.” in Materials and Methods section.

The data in Figures 4 & 5 and Figures 6 & 7 can be combined. The graphs present similar trends; why are both analyses needed? Some of this data can be moved to the ESI file.

Answer: We transferred Figures 4 and 5 to the ESI file (new Figures S3 and S4) and made Figures 6 and 7 the new Figures 4 and 5.

The fitted lines in Figures 5 and 7 are not overly representative of the raw data for some of the polymer samples. Error bars should be included.

Answer: The rate is calculated by averaging the difference between the averages over the number of days, so error bars cannot be produced.

The interaction between UVA and the PS sample is speculative. Can evidence/reference be provided.

Answer: Reference 25 suggested the presence of interaction. We added the Reference 25 in the corresponding sentence end as follows: “ The positive deviation means that there is no intermolecular interaction between the PP and UVA, while the negative value for PS does that there is one between them [28].”

There are two different references labelled as Reference 1. This makes it difficult to determine if the in-text references are correct/relevant.

Answer: We checked it, corrected the mistake.

References 10 and 11 are the same reference.

Answer: We checked it, corrected the mistake.

Figure S2 – The linear regression lines extend beyond the presented data for PP and PS samples. There is no indication in the Methods and Materials section that PP samples were prepared with 10-phr UVA.

Answer: We revised Figure S2. We did not prepared PP sample with 10-phr UVA. The sample containing 10-phr UVA was only PS.

Figure S3 shows the changes in MW distribution for PP samples degraded in pure water and sea water. A procedure for determining MW should be included.

Answer: The MW was determined by gel permeation chromatography (GPC) analysis. We added the GPC procedure in Materials and Methods.

Minor issues that need to be addressed:

 Line 91 – Define the abbreviation phr.

Answer: phr: per hundred resin .We defined it in Text.

Line 95 – Confirm that these are the correct references. Ref 16 seems more appropriate than Ref 26.

Answer: We added Ref 16 (new number 22). We have also left Ref 26 (new number 33) as believe it is an appropriate reference.

Line 126 – “Samples were placed in dried oven...” should be clarified. Drying oven?

Answer: Yes, we mistook it. We revised it!

Line 177-178 – “…can also degrade the crystalline it.” Must be corrected. Crystallinity?

Answer: It is “ crystalline part”. We rewrote it. We apologize for the confusion.

Line 187 – “ If there is no degradation of UVA due to degradation...” This statement reads awkwardly.

Answer: We revised the sentences as follows: The equilibrium of UVA bleaching from the polymer matrix can be explained by a physicochemical model if there is no degradation.

Line 201 – “as pronounced its increase” would be better stated as “as pronounced an increase”

Answer: We revised the sentence.

Comments on the Quality of English Language:

There are only minor grammatical mistakes. See comments regarding lines 126, 177-178, 187, and 201. There are other issues that affect the readability of the manuscript such as tense shifts, incorrect verb forms, verb agreement, etc.

Answer: We revised them. Thank you!

Reviewer 2 Report

Comments and Suggestions for Authors

Quality-changed behavior of ultraviolet light absorber (UVA: UV-326) in polymers over time during degradation was studied with an enhanced degradation method using sulfate ion radical in seawater. 

1.        The authors claim enhanced degradation method (EDM) is a better method to analyze the formation of microplastics by using the UV absorber (UVA). UVA was introduced into the polymers by thermal blending. How did the authors know the blending is uniform and would not influence the degradation? The authors should provide more evidence and references for using EDM to investigate the formation of microplastics.

2.        It is not proper to put figures in the middle of a paragraph. Please revise it.

3.        Figure 2 and 3 are the morphology changes of PP and PS, respectively. What about the other polymers? The authors claimed that whitening results from the chemical crystallization with no evidence. DSC or XRD may be needed to illustrate the degree of crystallization change.

4.        GPC analysis shows that the molecular weight of PP was degraded after EDM treatment. After the EDM treatment, how about the Mw change of other polymer samples (HDPE, LDPE, PLA, and PS)? 

5.        The degradation of UVA in different polymer samples was shown in the results, while the degradation of polymerswas not. The authors should provide more direct evidence of the polymer degradation.

6.        Figure 4 and 5 basically tell us the same information. It is confusing that both figures were contained in the manuscript. Figures 6 and 7 are the same.

7.        L190, the authors referred to a report on the leaching behaviors of UV stabilizers and claimed that it is a positive deviation from Raoults law for LDPE and a negative one for PS. What about the other polymers?

8.        From L216 to the very end, the order of the figures was miswritten. Please check and correct.

Comments on the Quality of English Language

 Extensive editing of English language required

Author Response

Reviewer 2

Comments and Suggestions for Authors

Quality-changed behavior of ultraviolet light absorber (UVA: UV-326) in polymers over time during degradation was studied with an enhanced degradation method using sulfate ion radical in seawater. 

  1. The authors claim enhanced degradation method (EDM) is a better method to analyze the formation of microplastics by using the UV absorber (UVA). UVA was introduced into the polymers by thermal blending. How did the authors know the blending is uniform and would not influence the degradation? The authors should provide more evidence and references for using EDM to investigate the formation of microplastics.

Answer: The polymers we used in this study, with the exception of PLA, are thermoplastic commodity plastics. The polymer and UVA blends were prepared by the melting mixer. This method of introducing UVA is called the melt-mixing (knead) method and has been commonly used in polymer processing. PP, LDPE, and HDPE are insoluble in common organic solvents and only slightly soluble at high temperatures in aromatic solvents such as xylene. Therefore, UVA has been commercially introduced into these polymers by the melt-mixing method. In this study, the UV-326 we used is UVA that has been used commercially for PP, PE and PS. Naturally, it is assumed to be added to these polymers by the melt-mixing method, and UV-326 is highly stable. No decomposition occurs at 180°C used for the melt-mixing. It is not clear whether UVA is dispersed in the polymer medium in a strict sense, but the reality is that blends are commercially produced in this manner. We do not see any other way to blend the UVA into these polymers.

Although methods for producing microplastics in a short period of time are currently being actively studied, our EDM method is the fastest method available. Unfortunately, there are no references other than ours.

  1. It is not proper to put figures in the middle of a paragraph. Please revise it.

Answer: We revised it.

  1. Figure 2 and 3 are the morphology changes of PP and PS, respectively. What about the other polymers? The authors claimed that whitening results from the chemical crystallization with no evidence. DSC or XRD may be needed to illustrate the degree of crystallization change.

Answer: We added the sentences and new reference as follows: It is well-known that the morphology of semi-crystalline polymers such as PP and HDPE is modified by chemi-crystallization during degradation [26]. The crystallinity increases as the degradation progresses, and the morphological change is accompanied by a change in appearance called whitening.

  1. GPC analysis shows that the molecular weight of PP was degraded after EDM treatment. After the EDM treatment, how about the Mw change of other polymer samples (HDPE, LDPE, PLA, and PS)? 

Answer: PS data also added. The decreasing behavior of Mw for the other polymers was also the same as those for PP and PS. In other words, with the EDM method, the MW decreased with time while maintaining the narrow distribution width.

  1. The degradation of UVA in different polymer samples was shown in the results, while the degradation of “polymers”was not. The authors should provide more direct evidence of the polymer degradation.

Answer: We added new Figures 7 and 8 to show the PP degradation.

  1. Figure 4 and 5 basically tell us the same information. It is confusing that both figures were contained in the manuscript. Figures 6 and 7 are the same.

Answer: We transferred Figures 4 and 5 to the ESI file (new Figures S3 and S4) and made Figures 6 and 7 as the new Figures 4 and 5.

  1. L190, the authors referred to a report on the leaching behaviors of UV stabilizers and claimed that it is a positive deviation from Raoult’s law for LDPE and a negative one for PS. What about the other polymers?

Answer: Other polymers (PP, HDPE and PLA) without other aromatic substituents also show the positive deviation, as LDPE.

  1. From L216 to the very end, the order of the figures was miswritten. Please check and correct.

Answer: We revised it.

Round 2

Reviewer 1 Report

Comments and Suggestions for Authors

Many of the suggestions and corrections have been made. However, there are still some minor issues that need to be addressed before this manuscript is ready for publication.

Some issues that need to be addressed:

The figure caption for Figure 7 appears to be incomplete.

Figure S1 - A UVA-free chromatogram of the polymer samples could be provided in the ESI.

Line 35 - Change comprises to comprised of.

Line 164 - This sentence is missing words.

Line 209 - .ClOH. needs to be corrected (ClOH) perhaps?

Comments on the Quality of English Language

There are many tense shifts within paragraphs, and number disagreements, and other minor grammatical errors that render this manuscript awkward to read. See lines 41-43 as an example.

Author Response

Many of the suggestions and corrections have been made. However, there are still some minor issues that need to be addressed before this manuscript is ready for publication.

Some issues that need to be addressed:

The figure caption for Figure 7 appears to be incomplete.

Answer: We revised the incomplete part.

Figure S1 - A UVA-free chromatogram of the polymer samples could be provided in the ESI.

Answer: We added the UVA-free chromatogram of the polymer sample in Figure S2.

Line 35 - Change comprises to comprised of.

Answer: We changed the word.

Line 164 - This sentence is missing words.

Answer: We added “of”.

Line 209 - .ClOH. needs to be corrected (ClOH) perhaps?

Answer: We corrected it.

Comments on the Quality of English Language

There are many tense shifts within paragraphs, and number disagreements, and other minor grammatical errors that render this manuscript awkward to read. See lines 41-43 as an example.

Answer: We revised them.

Reviewer 2 Report

Comments and Suggestions for Authors

All issue has been addressed

Author Response

Thank you very much for your review!